# ProTDyn: a foundation Protein language model for Thermodynamics and Dynamics generation

**Yikai Liu[1], Haoyang Zheng[1], Lining Mao[2], Yanbin Wang[1], Ming Chen[1], Guang Lin[1]***
[1]Purdue University    [2]Northwestern University

## Abstract

Molecular dynamics (MD) simulation has long been the principal computational tool for exploring protein conformational landscapes and dynamics, but its application is limited by high computational cost. We present ProTDyn, a foundation protein language model that unifies conformational ensemble generation and multi-timescale dynamics modeling within a single framework. Unlike prior approaches that treat these tasks separately, ProTDyn allows flexible independent and identically distributed (i.i.d.) ensemble sampling and dynamic trajectory simulation. Across diverse protein systems, ProTDyn yields thermodynamically consistent ensembles, faithfully reproduces dynamical properties over multiple timescales, and generalizes to proteins beyond its training data. It offers a scalable and efficient alternative to conventional MD simulations.

**Code:** `https://github.com/Harrydirk41/ProTDyn`

## 1 Introduction

Proteins are the fundamental building blocks of life, carrying out essential functions such as catalysis, signaling, and transport. Understanding their conformation flexibility and dynamic nature is crucial for uncovering the molecular mechanism of protein functions Whisstock & Lesk (2003). Traditional computational methods, such as molecular dynamics (MD) simulations, have been optimized and widely used for decades to study biomolecular processes such as protein folding and unfolding Shaw et al. (2010); Robustelli et al. (2018). However, MD remains computationally expensive, since the simulation time step must be orders of magnitude smaller than the timescales of biologically relevant processes Izaguirre et al. (1999). As a result, simulating long-timescale protein dynamics is often infeasible.

Recent advances in machine learning, particularly deep generative models, have introduced powerful alternatives to model proteins. Generative approaches have been developed to sample equilibrium conformational ensembles Lu et al. (2023); Lewis et al. (2025); Jing et al. (2024a); Noé et al. (2019); Zheng et al. (2024); Wayment-Steele et al. (2024) and, separately, to learn protein dynamics Jing et al. (2024b); Raja et al. (2025); Lelièvre et al. (2023); Du et al. (2024); Fu et al. (2022). Recent models also incorporate the principles of statistical mechanics that correlates thermodynamics and dynamics Raja et al. (2025); Arts et al. (2023).However, these models still rely on very small timesteps to propagate molecular dynamics, and generalize poorly beyond training dataset. Thus, scalable and transferable protein emulator models that can simultaneously describe both equilibrium conformational ensembles (thermodynamics) and conformational transitions across multiple timescales (dynamics) are still unavailable, which limits further progress such as accurately modeling protein biophysics, where both equilibrium ensembles and transition dynamics are essential for understanding protein–protein interaction, allosteric regulation, biocondensation, and conformational heterogeneity Brandsdal & Smalås (2000); Guo & Zhou (2016).

Based on recent progress and gaps, we introduce **ProTDyn**, a unified framework for generative modeling of Protein Thermodynamics and Dynamics. Unlike previous approaches that treat conformational ensemble generation and conformational dynamics propagation separately, ProTDyn

---

*Corresponding author: `guanglin@purdue.edu`

unifies them within a singlemulti-task architecture. ProTDyn is trained on hundreds of thousands of protein sequences and over a million of conformations, leveraging both single-structure and equilibrium MD simulation data to train the model for comprehensive understanding of protein conformational space. Moreover, ProTDyn can perform multi-timescale training which enables modeling conformational transitions of diverse protein systems with timescale ranging from nanoseconds to microseconds. This flexible scheduling bridges short- and long-timescale dynamics. Together, these capabilities are mutually reinforcing: accurate thermodynamic ensembles provide stable baselines for dynamic propagation, while realistic dynamics improve the diversity and fidelity of generated equilibrium ensembles.

In sum, we demonstrate three key capabilities of ProTDyn:

1. **Thermodynamics generation:** sampling independent and identically distributed (i.i.d.). equilibrium protein structures from the learned ensemble distributions that are consistent with Boltzmann statistics.

2. **Multiscale dynamics generation:** generating temporally coherent trajectories at multiple time-resolutions, capturing both fast local fluctuations and slow global transitions.

3. **Dynamics inpainting:** refining coarse time-resolution trajectories by recovering fine-grained time-resolution, physically plausible dynamic pathways.

By unifying thermodynamics and dynamics within a single generative framework, ProTDyn enables scalable, transferable, flexible, and computationally efficient protein modeling. In particular, compared to existing protein ensemble and dynamics generators, ProTDyn offers greater flexibility in generating conformational ensembles across diverse settings. We validate the accuracy and transferability of ProTDyn on multiple tasks, including generating conformational ensembles for proteins outside the training dataset and simulating protein dynamics beyond the training regime. Experimental results confirm that ProTDyn agrees well with reference MD simulations while generalizing effectively to unseen systems.

## 2 BACKGROUND

**Molecular dynamics.** Molecular dynamics (MD) simulates the time evolution of a molecular system by integrating Newton's equations of motion. For each particle $i$ in a molecular configuration $\mathbf{x} = (\mathbf{x}_1, \ldots, \mathbf{x}_N) \in \mathbb{R}^{3N}$, the equations are

$$M_i \ddot{\mathbf{x}}_i = -\nabla_{\mathbf{x}_i} U(\mathbf{x}_1, \ldots, \mathbf{x}_N), \tag{1}$$

where $M_i$ is the mass of particle $i$ and $U : \mathbb{R}^{3N} \to \mathbb{R}$ is the potential energy that is often modeled by a force field. By construction, an MD simulation at a fixed temperature $T$ will converge to the Boltzmann distribution of the system,

$$P(\mathbf{x}) \propto e^{-U(\mathbf{x})/k_B T}, \tag{2}$$

where $k_B$ is the Boltzmann constant.

**Deep generative modeling for proteins.** Recent advances in deep generative modeling have opened new opportunities to simulate protein conformational ensembles. These models can generate conformational ensembles and transitions within hours, offering an efficient alternative to conventional MD simulations. One line of work targets *thermodynamics*, directly learning the stationary Boltzmann distribution $P(\mathbf{x}|s)$ over conformations from structural databases or equilibrium MD trajectories Lu et al. (2023); Lewis et al. (2025); Jing et al. (2024a); Noé et al. (2019); Zheng et al. (2024); Wayment-Steele et al. (2024). Such models recover equilibrium ensembles but lack dynamical information. In addition to modeling thermodynamics, complementary approaches focus on *dynamics*, learning the transition density $P(\mathbf{x}_{t+\Delta t} \mid \mathbf{x}_t, s)$ to accelerate MD Jing et al. (2024b); Raja et al. (2025); Lelièvre et al. (2023); Du et al. (2024); Fu et al. (2022). Although these methods can predict short-time kinetics, they are often trained on limited MD data, restricting their ability to generalize to rare or long-timescale transitions. Despite rapid progress, current approaches remain specialized in either thermodynamics or dynamics, overlooking their intrinsic connection based on statistical mechanics. A unified foundation model capable of predicting both equilibrium ensembles and transition dynamics across scales has yet to be established.

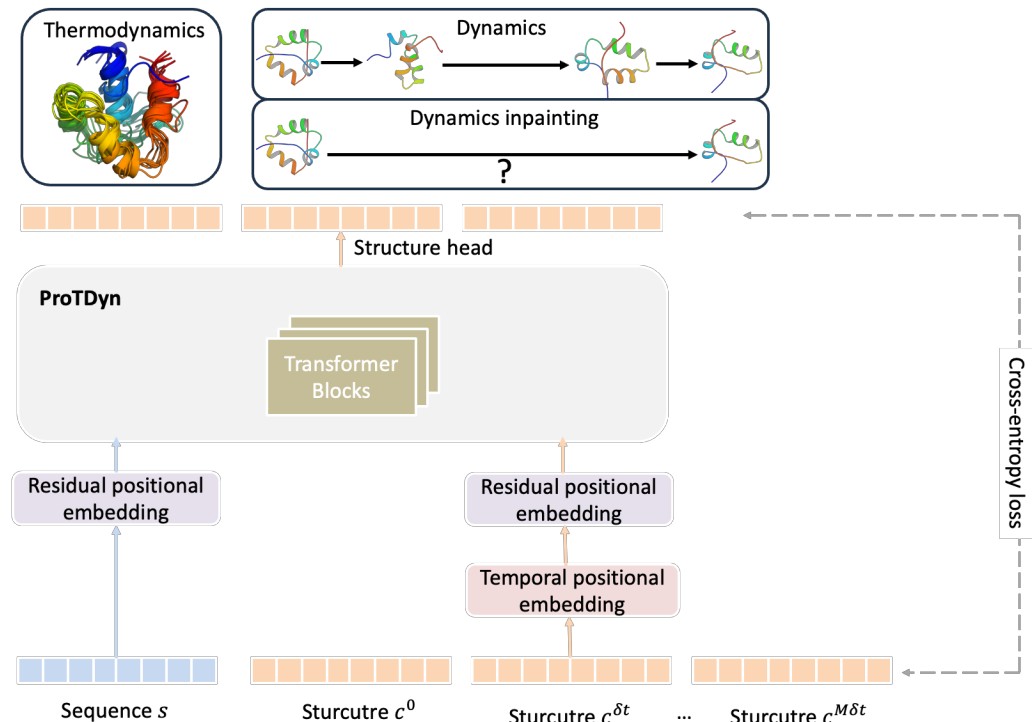

Figure 1: An illustrative framework of ProTDyn. ProTDyn is a protein language model that operates on discretized representations of protein sequence and structure. It leverages a powerful autoregressive transformer architecture to simultaneously perform three tasks: (i) equilibrium conformational ensemble generation (thermodynamics), (ii) forward trajectory generation across multiple timescales (dynamics), and (iii) recovery of fine-grained trajectories from coarse trajectories (dynamics inpainting).

**Protein conformation representation.** Although proteins are inherently three-dimensional objects, recent advances have shown that conformations can also be represented as sequences of discrete tokens, enabling the use of powerful sequence modeling techniques. In this work, we adopt the pretrained ESM3 Hayes et al. (2025) structure tokenizer to map protein conformations to tokenized sequences $\mathbf{c} \in \mathbb{Z}^N$, where each residue is assigned one of the $4{,}096$ structure tokens (plus $4$ special tokens). These tokens provide a compact, learned representation of the local structural neighborhood around each residue. The discretization is performed with a VQ-VAE encoder Van Den Oord et al. (2017), while a paired decoder reconstructs the generated token sequences back into three-dimensional coordinates.

## 3 METHOD

In this section, we will introduce the high-level framework of ProTDyn, as illustrated in Fig. 1. ProTDyn is a multi-task protein language model designed to perform three complementary tasks within a single framework: (i) equilibrium conformation ensemble generation (thermodynamics), (ii) multi-timescale dynamic trajectory generation (dynamics), and (iii) fine-grained trajectories recovery from coarse time-resolution trajectories (dynamics inpainting). The model operates by autoregressively predicting the structure tokens of successive residues. Its architecture builds upon the state-of-the-art protein language model ESM3, and additionally incorporates a temporal positional embedding to encode dynamic transition information.

### 3.1 OBJECTIVE

We represent protein conformations in a discrete tokenized space and define three complementary learning objectives:

1. **Thermodynamics:** the thermodynamics head of ProtDyn aims to learn the equilibrium distribution of conformations $P_\theta(\mathbf{c} \mid s)$. Specifically, it models the equilibrium conformation distributions autoregressively as

$$P_\theta(\mathbf{c} \mid s) = \prod_{i=0}^{N-1} P_\theta(\mathbf{c}_i \mid \mathbf{c}_{<i}, s), \tag{3}$$

where $\mathbf{c}_i$ denotes the structure token of residue $i$, and $\mathbf{c}_{<i}$ represents all preceding residues. The thermodynamics head is learned by minimizing the cross entropy between $P_\theta$ and the observed protein equilibrium conformation ensemble distribution:

$$\mathcal{L}_{\text{thermo}}(\theta) = -\mathbb{E}_{(s,\mathbf{c})\sim\mathcal{D}}\left[\sum_{i=0}^{N-1} \log P_\theta(\mathbf{c}_i \mid \mathbf{c}_{<i}, s)\right], \tag{4}$$

where $\mathcal{D}$ denotes the dataset of protein sequences and their equilibrium conformations.

2. **Dynamics:** the dynamics head aims to learn the temporal correlations across multiple timescales $P_\theta(\mathbf{C} \mid s)$, where $\mathbf{C} = (\mathbf{c}^0, \dots, \mathbf{c}^{M\delta t})$ is a trajectory segment of length $M$ with time step $\delta t$. We factorize the trajectory distribution as

$$P_\theta(\mathbf{C} \mid s) = \prod_{j=0}^{M} P_\theta(\mathbf{c}^{j\delta t} \mid \mathbf{C}^{<t}, s), \tag{5}$$

where $\mathbf{C}^{<t}$ represents all preceding conformations up to time step $t-1$. Each conformation $\mathbf{c}^{j\delta t} = (\mathbf{c}_0^{j\delta t}, \dots, \mathbf{c}_{N-1}^{j\delta t})$ is further decomposed residue-wise as

$$P_\theta(\mathbf{c}^{j\delta t} \mid \mathbf{C}^{<t}, s) = \prod_{i=0}^{N-1} P_\theta(\mathbf{c}_i^{j\delta t} \mid \mathbf{c}_{<i}^{j\delta t}, \mathbf{C}^{<t}, s). \tag{6}$$

The model parameters are optimized by minimizing the negative log-likelihood of observed trajectory data:

$$\mathcal{L}_{\text{dyn}}(\theta) = -\mathbb{E}_{(s,\mathbf{C})\sim\mathcal{D}}\left[\sum_{j=0}^{M} \log P_\theta(\mathbf{c}^{j\delta t} \mid \mathbf{C}^{<t}, s)\right], \tag{7}$$

where $\mathcal{D}$ denotes the dataset of protein sequences paired with dynamic segments of length $M$. In this project, we aim to capture protein dynamical behaviors at multiple timescales. Specifically, we let the model learn $\delta t = 1\,\text{ns}$, $10\,\text{ns}$, and $100\,\text{ns}$ resolution. Additionally, we set a memory kernel of $M = 10$, which corresponds to effective timescales of $10\,\text{ns}$, $100\,\text{ns}$, and $1000\,\text{ns}$, respectively.

3. **Dynamic inpainting:** The dynamics head enables trajectory generation with large timesteps, but this comes at the cost of losing fine-grained temporal resolution. To address this, we introduce a dynamic inpainting head that reconstructs physically plausible transition sequences between metastable conformational states. Formally, the inpainting task is to recover fine-resolution conformations trajectory $\mathbf{C}$ between two states $\mathbf{c}^0$ and $\mathbf{c}^{M\delta t}$, modeled from coarse trajectories $P_\theta(\mathbf{C} \mid \mathbf{c}^0, \mathbf{c}^{M\delta t}, s)$. Similar to dynamics head, the dynamic inpainting between two states $\mathbf{c}^0$ and $\mathbf{c}^{M\delta t}$ is formulated as an autoregressive conditional generation problem:

$$P_\theta(\mathbf{C} \mid \mathbf{c}^0, \mathbf{c}^{M\delta t}, s) = \prod_{j=1}^{M-1} P_\theta(\mathbf{c}^{j\delta t} \mid \mathbf{C}^{<t}, \mathbf{c}^0, \mathbf{c}^{M\delta t}, s), \tag{8}$$

and the training objective is:

$$\mathcal{L}_{\text{dynI}}(\theta) = -\mathbb{E}_{(s,\mathbf{C})\sim\mathcal{D}}\left[\sum_{j=1}^{M-1} \log P_\theta(\mathbf{c}^{j\delta t} \mid \mathbf{C}^{<t}, \mathbf{c}^0, \mathbf{c}^{M\delta t}, s)\right], \tag{9}$$

The choice of time step is 1 ns and 10 ns and memory kernel follows the same as training of dynamics head.

Together, these objectives enable the model to simultaneously generate equilibrium ensembles and reproduce protein dynamics at multiple timescales. During training, ProTDyn is optimized to minimize losses from all three heads. The losses are combined using hyperparameter weights:

$$\mathcal{L}_{\text{ProTDyn}} = \omega_1 \mathcal{L}_{\text{thermo}} + \omega_2 \mathcal{L}_{\text{dyn}} + \omega_3 \mathcal{L}_{\text{dynI}}. \tag{10}$$

where $\omega_1$, $\omega_2$, and $\omega_3$ are hyperparameters.

## 3.2 TRANSFORMER ARCHITECTURE

We adopt a transformer backbone following ESM3 Hayes et al. (2025). In particular, we use Pre-LN instead of Post-LN, rotary positional embeddings (RoPE) Su et al. (2024) instead of absolute positional embeddings, and SwiGLU activations instead of ReLU. To encode both residue- and temporal-level positional information, we introduce a two rotary embedding scheme.The residue-embedding layer encodes positions along the protein sequence by assigning integer indices to sequential residues (1 for the first residue, 2 for the second, and so on). Residue-embeddings are applied to both sequence tokens and structure tokens. To incorporate temporal information for modeling multiscale dynamics, we introduce an additional temporal-embedding layer. The temporal embedding is defined such that the smallest temporal unit corresponds to 1 ns, which is the finest time resolution accessible to our model. For structure tokens, all tokens in the first structural segment are assigned a temporal value of 0, those in the second segment are assigned $\delta t$, those in the third segment are assigned $2\delta t$, and so forth. The temporal-embedding layer is applied to structure tokens before the residue-embedding layer, and is not applied to sequence tokens. The training data incorporates $\delta t = 1, 10, 100$ ns. At inference time, any integer timestep between 1 and 100 ns will work.

## 4 EXPERIMENT

In this section, we describe the details of our training and evaluation setup. Our results demonstrate that ProTDyn achieves performance equivalent to, or even surpassing, state-of-the-art protein ensemble generators in equilibrium ensemble generation, while additionally capable of capturing long-timescale dynamics.To demonstrate the effectiveness of unifying the thermodynamics and dynamics heads within a single model, we additionally train a model with same architecture and model parameters using only the dynamics information. We show that the unified model achieves significantly better performance on all dynamic prediction tasks, indicating that incorporating thermodynamic signals provides complementary information that improves dynamic inference.

## 4.1 TRAINING SETUP

**Data** Following BioEmu Lewis et al. (2025), we combine single-structure and equilibrium MD datasets for thermodynamics training. For the single-structure dataset, we use the Swiss-Prot subset from the AlphaFold database, which contains 542,378 sequence–structure pairs. For equilibrium MD simulations, we include two sources: mdCath and BioEmu. For dataset construction and simulation details, we encourage readers to read both sources for a comprehensive understanding.

- **mdCath** Mirarchi et al. (2024): 5,398 proteins with simulation lengths up to 500 ns.
- **BioEmu** Lewis et al. (2025): prolonged simulations of diverse protein systems. The BioEmu corpus includes several subsets:
  - **Octapeptides** Charron et al. (2025): 1,100 peptides of length 8, each simulated for 5 $\mu$s.
  - **CATH1** Charron et al. (2025); Sillitoe et al. (2021): 50 CATH domains, each simulated for 100 $\mu$s.
  - **CATH2** Sillitoe et al. (2021): 1,100 CATH domains, each simulated for 39 $\mu$s.
  - **MEGAsim** Tsuboyama et al. (2023): extended simulations of 271 wild-type proteins, together with 1 $\mu$s simulations for each of 22,118 single-point mutants.

For dynamics and dynamics inpainting training, we use simulations from mdCath at temporal resolutions of 1 ns, and from the Octapeptides, CATH2, and MEGAsim subsets at temporal resolutions of 10 and 100 ns.

**Model details** For better sequence and structure representation, we use the pretrained sequence and structure embedding head from ESM3 and freeze it throughout the training. The backbone of ProTDyn consists of 24 transformer blocks, which together contain 1.4 billion parameters.

**Optimization** We trained the model using AdamW optimizer Loshchilov & Hutter (2017) with a learning rate of $4 \times 10^{-4}$ and weight decay of $1 \times 10^{-5}$. We used a learning rate scheduler that reduces the learning rate by a factor of 0.5 if the training loss does not improve for 5 consecutive epochs.

## 4.2 INFERENCE SETUP

We conduct ablation studies with three types of sampling methods:

1. **Thermodynamics**: independent and identically distributed (i.i.d.) sampling of protein conformational ensembles. In our experiments, we generated 2,000 ensembles for each protein.

2. **Dynamics (10 ns)**: forward generative simulations with a time step of 10 ns. In our experiments, we generated 50 independent trajectories, each propagated for 100 steps, corresponding to a total of $50~\mu$s of simulation time and 5,000 conformational ensembles.

3. **Dynamics (100 ns)**: Forward generative simulations with a coarse time interval of 100 ns. To recover fine-grained details, each 100 ns interval is refined by dynamics inpainting into ten 10 ns sub-intervals. Similarly to the "Dynamics (10 ns)" sampling head, we generate 50 independent trajectories, each propagated for 10 coarse steps and refined by inpainting, also yielding $50~\mu$s of simulation time and 5,000 ensembles.

For the experiment of the **thermodynamics head**, we generate ensembles for 50 proteins from the CATH1 dataset. These proteins are included in the thermodynamic training set and serve primarily as a benchmark against the baseline model. To further assess generalization, we also generate ensembles for 10 octapeptides that were not included in the training dataset.

For the experiment of the **dynamics head**, we apply both dynamic sampling heads to 50 proteins from the CATH1 dataset. Although these proteins are part of the thermodynamics training set, they are excluded from dynamic and inpainting training. This setting enables a direct evaluation of the generalization capacity of ProTDyn's dynamic generation. We conduct same dynamic experiments using the model trained with only dynamic data, and use the same initial structures as starting point of dynamics during inference.

## 4.3 EVALUATION

**Distributional similarity** Our main comparison of ensemble distribution quality is against the BioEmu model Lewis et al. (2025). We selected BioEmu as our benchmark baseline because it represents the most recent major advance in quantitative protein conformation ensemble modeling. In contrast, previous approaches were typically trained on very limited MD simulations of short duration. In fairness, we used a training dataset that closely matches the one used by BioEmu. We report the Jensen-Shannon divergence (JSD) between the reference MD simulations and predicted conformation ensembles along various collective variables (CVs). The first set of CVs are low dimensional physical features: radius of gyration (Rg) and root mean square distance (RMSD) w.r.t the native structure predicted by AlphaFold 3 Abramson et al. (2024). The second set of CVs are the top 2 independent components from time-lagged independent components analysis (TICA) Schultze & Grubmuuller (2021), representing the slowest dynamic modes of proteins. We evaluate distributional similarity on ensembles generated by all three sampling heads.

**Dynamical content** We first evaluate the autocorrelation functions of the TICA components. To further assess the accuracy of predicted dynamics, we discretize the conformational space into states and construct Markov state models (MSMs) Husic & Pande (2018); Chodera & Noé (2014); Pande et al. (2010); Bowman et al. (2013) to estimate transition probabilities and stationary distributions.

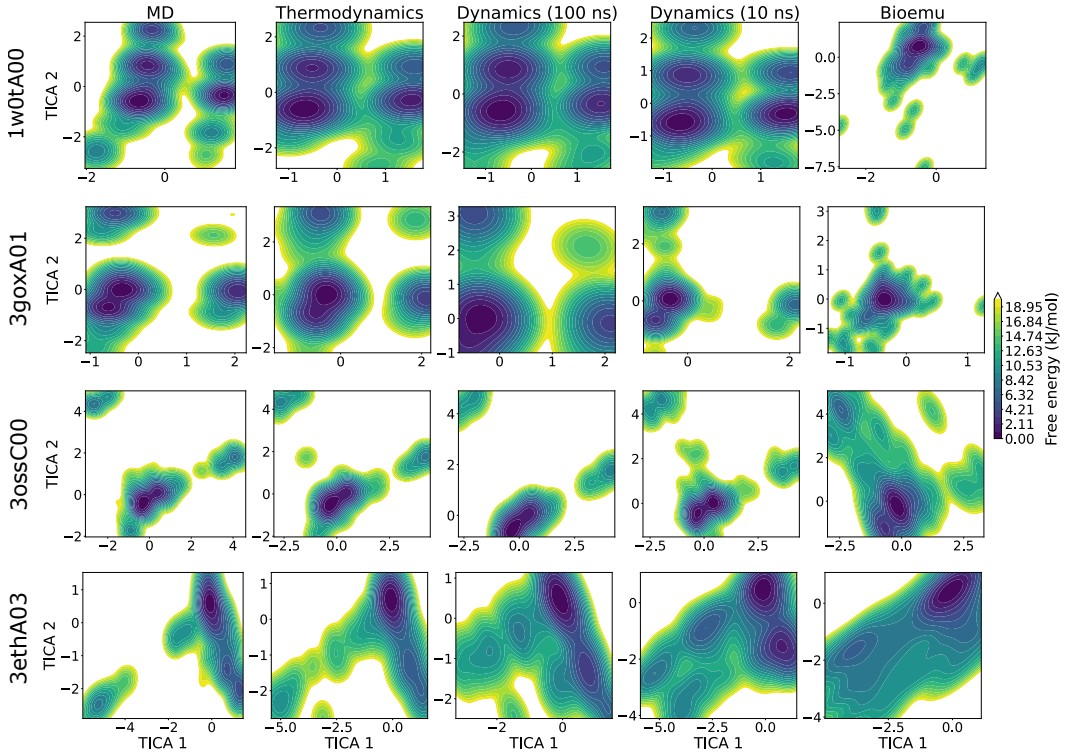

Figure 2: Free energy surface along the top two TICA components, parameterized from the backbone torsion angles of reference MD simulations. The TICA projection is then applied to conformational ensembles generated by the three sampling heads of ProTDyn: (1) "Thermodynamics", (2) "Dynamics (100 ns)", and (3) "Dynamics (10 ns)", as well as to ensembles generated by the baseline model BioEmu.

MSMs built from reference 100 $\mu$s MD trajectories serve as ground truth, while the MSMs constructed from the ProTDyn trajectories are used for evaluation. For direct comparison, we also construct MSMs from the first 25%, 50%, and 75% of the reference MD trajectories, corresponding to 25, 50, and 75 $\mu$s of simulation time, respectively. Following metrics from previous works Jing et al. (2024b); Raja et al. (2025), we report the Jensen–Shannon divergence (JSD) between the stationary distributions of Markov states obtained from ProTDyn and those from reference MD. In addition, we report the JSD between the corresponding transition probability matrices. We also sample dynamic trajectories from MSM, and compute the mean negative log-likelihood (NLL) of dynamic trajectories. (details of MSM construction and evaluation methods in Appendix A)

### 4.4 RESULT

**Distributional similarity result** We visualize the free energy surface (FES) of the TICA projection in Fig. 2. As shown, all three ProTDyn sampling heads recover the major conformational metastable states along with qualitatively correct energy barriers that separate them.

For quantitative assessment, we report the JSD of the conformation ensembles generated by ProTDyn using the three sampling heads and BioEmu, average over all proteins in the test data in Table 1. Across all collective variables, ProTDyn shows good agreement with the reference 100 $\mu$s simulations MD distributions. Among the three sampling heads, "Thermodynamics" sampling yields the highest-quality ensembles. This is expected, as i.i.d. samples are directly generated from the equilibrium distribution and while the invariant probably from dynamics head may be affected by possible error accumulation in autoregressive dynamic simulations.

Interestingly, we observe that "Dynamics (100 ns)" produces higher quality ensembles than "Dynamics (10 ns)". We hypothesize that the "Dynamics (10 ns)" head, which requires many successive

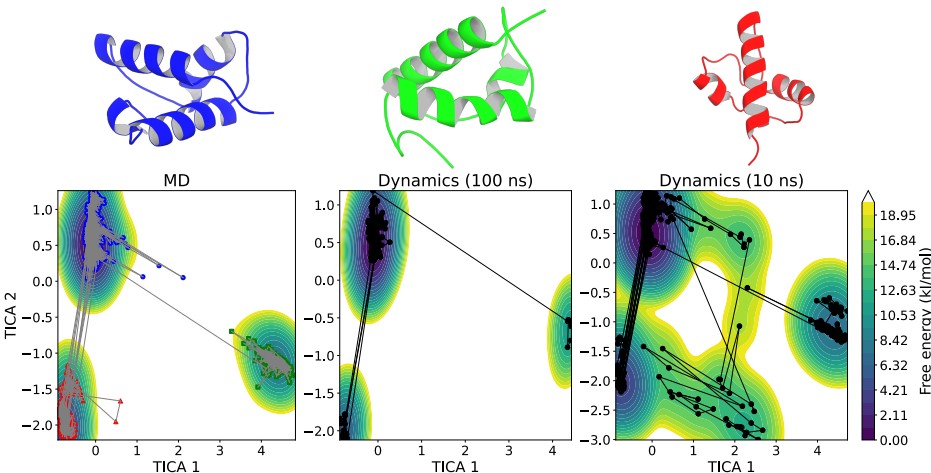

Figure 3: Representative conformational metastable states and dynamic transition pathways illustrated on the 2D TICA free energy surface (FES) for a CATH1 protein system: 1b43A02.

steps to reach long timescales, is more susceptible to error accumulation and degradation of predictive accuracy. In contrast, the "Dynamics (100 ns)" head provides a more stable strategy: large time steps enable robust exploration of long-timescale conformational transitions, while dynamic inpainting subsequently refines each interval to recover fine-grained transition details.

Another significant observation is that without training on thermodynamic information, the dynamics-only model diverges rapidly from the ground truth. It fails to maintain the folded structure and performs substantially much worse in capturing the distribution of collective variables. This demonstrates the importance of integrating thermodynamic signals to predict the dynamic evolution.

Finally, we evaluate the generalization of the "Thermodynamics" head on 10 octapeptides unseen in the training dataset, with results reported in Table 2. We observe a high level of agreement with the reference MD simulations and thermodynamic performance that is very close to the baseline BioEmu model. Note that these 10 proteins are included in the BioEmu training dataset but not for ProTDyn. This demonstrates the strong generalizability of ProTDyn's thermodynamic head beyond its training dataset.

Table 1: Distributional similarity evaluation on CATH1 test dataset. Metrics are reported as Jensen–Shannon divergence (JSD) over the radius of gyration (Rg), root-mean-square distance (RMSD) w.r.t the native structure, and the top two TICA components capturing slow protein dynamics. Evaluation is conducted for ProTDyn under three different sampling heads and for the baseline method BioEmu.

| Model | Rg $\downarrow$ | RMSD $\downarrow$ | TICA $\downarrow$ |
|---|---|---|---|
| **ProTDyn**-Thermodynamics | **0.023** | **0.012** | **0.155** |
| **ProTDyn**-Dynamics (10 ns) | 0.052 | 0.032 | 0.278 |
| **ProTDyn**-Dynamics (100 ns) | 0.030 | 0.018 | 0.206 |
| **ProTDyn-dynamics-only** (100 ns) | 0.077 | 0.142 | 0.315 |
| BioEmu | 0.082 | 0.137 | 0.293 |

**Dynamical content result** We visualize the dynamic pathways of a test protein in CATH1 in Fig. 3, where systems demonstrate multiple conformation states. Both ProTDyn dynamic sampling heads can recover the conformation ensembles and the rare transition events. For qualitative analysis, we visualize the autocorrelation of the first two TICA components over 800 ns across all test proteins in Fig. 4. The autocorrelation functions from "Dynamics (100 ns)" sampling head closely reproduce the reference results, whereas the autocorrelation functions from "Dynamics (10 ns)" head decay much faster than the the reference results. This observation is consistent with our expectation that

Table 2: Distributional similarity evaluation on the Octapeptide test dataset, using the same evaluation metrics as in CATH1 test dataset. Note that the these proteins are used as training proteins for the baseline model Bioemu but not for our ProTDyn model.

| Model | Rg ↓ | RMSD ↓ | TICA ↓ |
|---|---|---|---|
| **ProTDyn**-Thermodynamics | 0.034 | 0.020 | 0.207 |
| BioEmu | **0.031** | 0.020 | **0.134** |

smaller time intervals accumulate errors more rapidly, leading to a loss of long-timescale correlations. Finally, we construct MSM and quantitatively evaluate the JSD of stationary distributions and transition probability from MSM and the negative log-likelihood of dynamic paths between full reference MD simulations, ProTDyn-generated trajectories, and different portions of the reference MD data (Table 3). The 50 $\mu s$ trajectory generated with the "Dynamics (10 ns)" head recovers MSM stationary distributions and transition probabilities with accuracy comparable to 25 $\mu s$ of MD, while the "Dynamics (100 ns)" head achieves MSM quality similar to 50 $\mu s$ of MD. These results demonstrate the strong dynamical fidelity of ProTDyn and highlight the effectiveness of combining large-timestep generation with inpainting to recover fine-scale transitions.Finally, evaluation of the dynamic trajectories generated by the dynamics-only model demonstrates that the absence of thermodynamic supervision results in poor dynamic performance, producing unrealistic trajectories and failing to recover realistic stationary distributions and transition probabilities. These evaluations highlight the necessity of unifying thermodynamics and dynamics head.

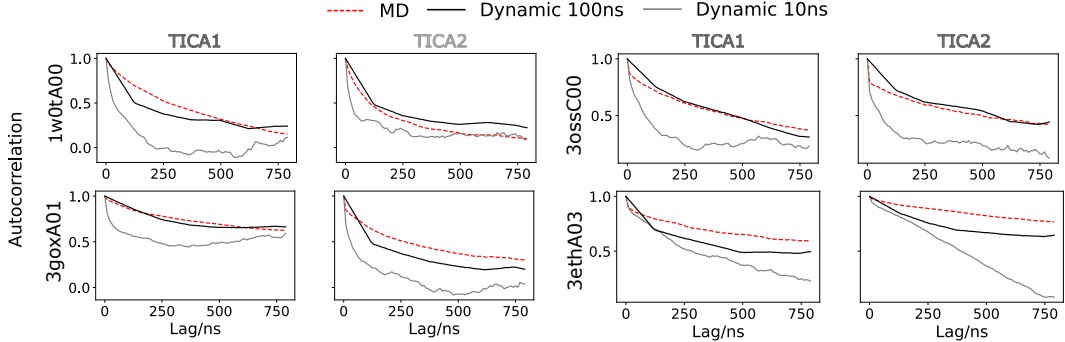

Figure 4: Autocorrelation of the top two TICA components from 0 to 800 ns lag time, evaluated on four test CATH1 proteins using reference MD trajectories and dynamic trajectories generated by the two dynamic sampling heads of ProTDyn.

Table 3: Dynamical contents evaluation of **ProTDyn** under two dynamic sampling heads. Evaluation metrics include Jensen-Shannon (JS) divergence of the stationary distribution of and transition probability between Markov states, and mean negative log-likelihood (NLL) of 200 ns transition paths.

| Model | Stationary JSD ↓ | Transition JSD ↓ | NLL ↓ |
|---|---|---|---|
| **ProTDyn**-Dynamics (10 ns) | 0.042 | 0.105 | 0.806 |
| **ProTDyn**-Dynamics (100 ns) | 0.021 | 0.055 | 0.656 |
| **ProTDyn-dynamics-only** (100 ns) | 0.088 | 0.167 | 1.013 |
| 25 $\mu s$ MD | 0.040 | 0.117 | 0.815 |
| 50 $\mu s$ MD | 0.018 | 0.049 | 0.682 |
| 75 $\mu s$ MD | 0.007 | 0.019 | 0.639 |

## 5 DISCUSSIONS

**Limitations** Similar to other generative models, ProTDyn is constrained by the availability and scale of training data, especially equilibrium MD simulation data. Its performance could be improved with access to larger and more diverse datasets. In addition, the current training procedure manually specifies the memory kernel at each timescale, without a rigorous investigation of its optimal form. Recent studies have explored principled approaches to designing or learning memory kernels, and incorporating these insights could further strengthen the model. Wu et al. (2024); Ge et al. (2024)

**Opportunities** Two important capabilities of ProTDyn that we have not explored in detail are transition path sampling and likelihood evaluation. Currently, ProTDyn employs transition path sampling purely as an inpainting technique to recover short-timescale details within long-timescale generative dynamics trajectories. However, rigorous transition path sampling requires specialized sampling strategies such as transition interface sampling, milestoning, and string methods Van Erp & Bolhuis (2005); Pan et al. (2008); Votapka & Amaro (2015), and general-purpose MD simulations are not the most suitable training data for this purpose.

A key distinction of ProTDyn is that, through autoregressive modeling of discrete protein tokens, it provides *exact* likelihood evaluation for both conformational ensembles and dynamic trajectories. This contrasts with popular diffusion- or flow-based protein ensemble generators, where likelihoods are only approximated and are often computationally prohibitive to evaluate Song et al. (2020). Exact likelihoods create opportunities for integration with physics-based energy functions, or even the development of a new top-down protein force field.

Finally, current generative models for proteins are not explicitly grounded in principles in statistical mechanics, such as detailed balance Tolman (1925). With its likelihood evaluation capability, ProTDyn offers an opportunity to enforce such physical laws, thereby unifying thermodynamics and dynamics. This could help overcome the limitations of data scarcity and enable more physically consistent generative modeling.

## ETHICS STATEMENT

We have read and adhered to the ICLR Code of Ethics. Our study does not involve human subjects, sensitive personal data, or applications with direct negative societal consequences. All experiments are conducted on publicly available datasets and open-source models. We are committed to fairness, transparency, and responsible reporting of our results. To the best of our knowledge, this work does not present risks of misuse beyond those already inherent in large language models.

## REPRODUCIBILITY STATEMENT

We have made significant efforts to ensure reproducibility of our results. Details of the evaluation protocols are provided in the Appendix. All datasets used are publicly available, and the preprocessing steps are clearly documented.

## ACKNOWLEDGEMENTS

Guang Lin would like to thank the support of National Science Foundation (DMS-2533878, DMS-2053746, DMS-2134209, ECCS-2328241, CBET-2347401 and OAC-2311848), and U.S. Department of Energy (DOE) Office of Science Advanced Scientific Computing Research program DE-SC0023161, the SciDAC LEADS Institute, and DOE–Fusion Energy Science, under grant number: DE-SC0024583. Ming Chen would like to thank supports from the National Science Foundation (CSEDI EAR-2246687).

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

# A EXPERIMENT DETAILS

## A.1 DISTRIBUTIONAL SIMILARITY EVALUATION

**Radius of gyration.** The radius of gyration ($R_g$) is a fundamental descriptor of protein structure, quantifying the overall compactness of a molecule. It is defined as the root-mean-square distance of the constituent atoms from their common center of mass, thereby capturing how mass is distributed around the protein's centroid.

The radius of gyration is defined as

$$R_g = \sqrt{\frac{1}{N}\sum_{i=1}^{N}\left\|\mathbf{r}_i - \mathbf{r}_{\mathrm{cm}}\right\|^2}, \tag{11}$$

where $N$ is the number of atoms, $\mathbf{r}_i$ is the position vector of the $i$-th atom, and $\mathbf{r}_{\mathrm{cm}}$ is the position vector of the center of mass. This expression measures the spatial dispersion of the atomic positions relative to the center of mass. In this work, $R_g$ is computed using only the backbone atoms.

**Root Mean Square Distance (RMSD).** The root mean square distance (RMSD) is a standard metric in structural biology for quantifying the similarity between two protein conformations. RMSD is computed by superimposing the two structures and calculating the square root of the average squared distance between corresponding atoms:

$$\mathrm{RMSD} = \min_{T_g \in \mathrm{SE}(3)} \sqrt{\frac{1}{N}\sum_{i=1}^{N}\left\|T_g(\mathbf{r}_i) - \mathbf{r}_i^{\mathrm{ref}}\right\|^2}, \tag{12}$$

where $N$ is the number of atoms, $\mathbf{r}_i$ are the atomic coordinates of the structure under comparison, $\mathbf{r}_i^{\mathrm{ref}}$ are the coordinates of the reference structure, and $T_g \in \mathrm{SE}(3)$ denotes the optimal rigid-body transformation (rotation and translation) aligning the two structures. Lower RMSD values indicate greater structural similarity.

**Time-lagged Independent Component Analysis (TICA).** TICA is a linear method for extracting the slowest dynamical modes from time-series data, widely used in molecular dynamics. Unlike PCA, which finds directions of largest variance, TICA identifies directions with maximal autocorrelation at lag time $\tau$. It solves the generalized eigenvalue problem

$$C_\tau r_i = C_0 \lambda_i r_i, \tag{13}$$

where $C_0$ is the covariance matrix, $C_\tau$ is the time-lagged covariance, and $\lambda_i$ are the time-autocorrelations. The resulting components capture the slow collective motions that dominate long-timescale protein dynamics. In this project, we choose the feature as protein backbone torsion angles, and extract the top 2 TIC components.

## A.2 DYNAMICAL CONTENTS EVALUATION

**Autocorrelation** We define the autocorrelation of each TICA component as

$$\mathbb{E}\big[(y_t - \mu)(y_{t+\delta t} - \mu)\big]/\sigma^2 \tag{14}$$

where $\mu, \sigma$ are computed from the reference MD simulation trajectory. We conduct TICA on lag times $\delta t = [10, 20, \dots, 800]$ ns.

**Markov state models** Markov state models (MSMs) provide a statistical framework for describing the long-timescale dynamics of biomolecules by coarse-graining the continuous conformational space into a finite set of metastable states. The dynamics are modeled as a discrete-time Markov chain, where the probability of transitioning between states depends only on the current state and a chosen lag time $\tau$. Formally, the state-to-state transition probabilities are encoded in a transition matrix $T(\tau)$:

$$p_j(t + \tau) = \sum_i p_i(t)\, T_{ij}(\tau), \tag{15}$$

where $p_i(t)$ is the probability of being in state $i$ at time $t$, and $T_{ij}(\tau)$ is the probability of transitioning from state $i$ to $j$ over lag time $\tau$. At equilibrium, MSMs satisfy the stationary distribution condition

$$\boldsymbol{\pi} = \boldsymbol{\pi} T(\tau), \quad \sum_i \pi_i = 1, \tag{16}$$

where $\pi_i$ denotes the equilibrium probability of state $i$.

We follow previous works Jing et al. (2024b); Raja et al. (2025) and build MSM using Deeptime Hoffmann et al. (2021). We first represent protein systems with backbone torsion angles and run TICA to obtain the top 2 Time Independent Component dimensions. We then perform k-means clustering of the reference MD simulations into 10 clusters. We then fit a MSM with a lag time of 10 ns. This gives us the transition probability matrix $T$. The stationary distribution $\pi$ can be easily obtained as the left eigenvector of the transition matrix with eigenvalue 1. We use the same settings to construct MSM for reference MD simulations, dynamic trajectories generated by ProTDyn, and different portions of reference MD simulations.

We evaluate the following two metrics:

- **Stationary distribution Jenson-Shannon divergence** We evaluate the stationary distribution of each MSM state for reference and comparison trajectories, and compute the JSD between the categorical distributions.

- **Transition probability Jensen–Shannon divergence** We evaluate the transition probability matrix between MSM states for both reference and comparison trajectories, and compute the JSD between the corresponding transition probability distributions.

- **Trajectory negative log likelihood** We use the stationary distribution of reference MSM to generate 1000 initial states: $s_0 \sim \pi_{\text{ref}}$, and then propagate a dynamic path of length $L = 20$ from each iniitial state for each comparison MSM using its transition proability matrxi: $s_{t+1} \sim T_{s_t}$. The probability of a path can thus be written as $P(s_1, ..., s_L) = -\frac{1}{L} \sum_{t=1}^{L} \log T_{\text{ref}_{s_t, s_{t+1}}}$. We report the mean negative log likelihood over all trajectories.

## B  ADDITIONAL EXPERIMENT RESULTS

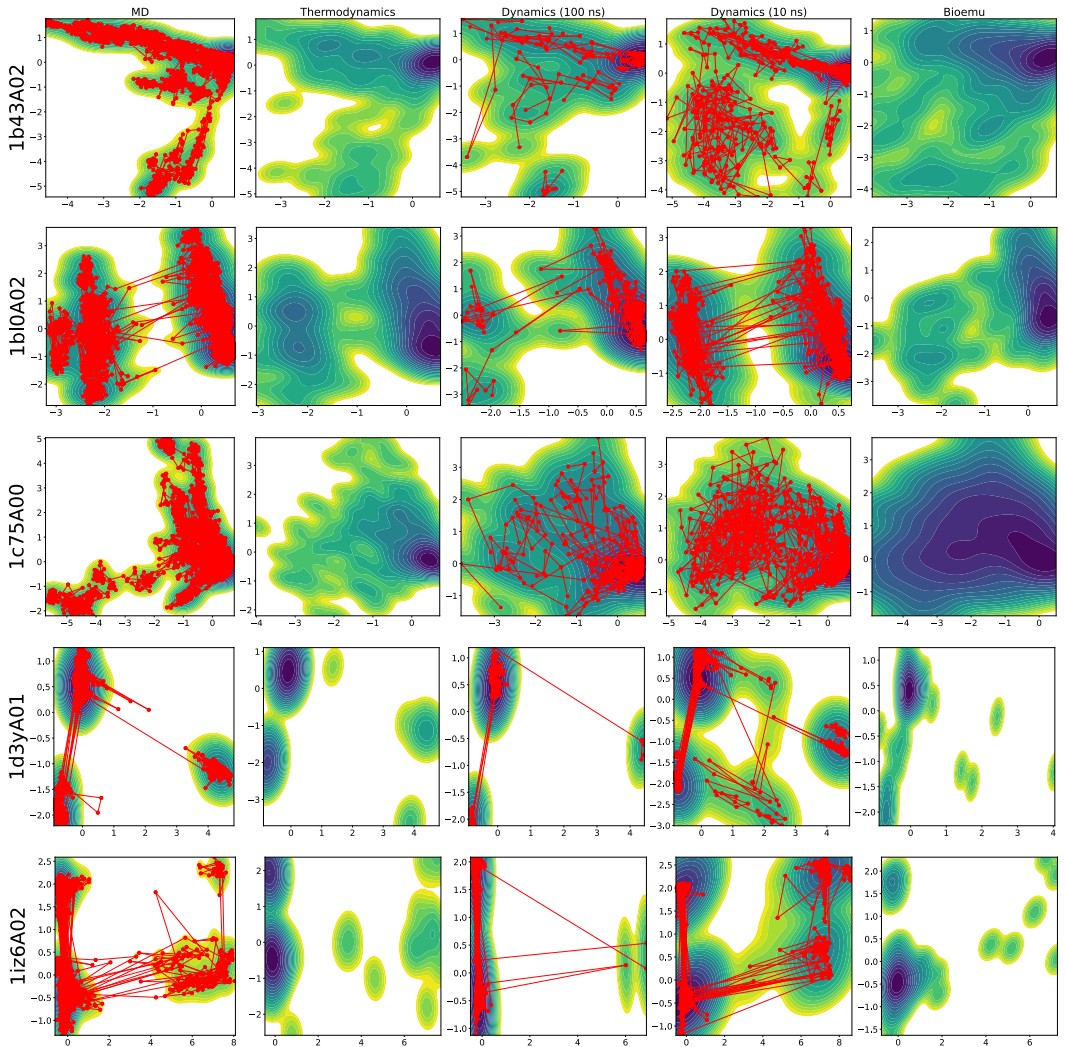

Figure 5: Additional results on 5 test CATH1 proteins: free energy surface along the top two TICA components and dynamic transition pathways. Note that the thermodynamics head of ProTDyn and baseline model Bioemu are both i.i.d sampler and thus does not have a dynamic trajectory.

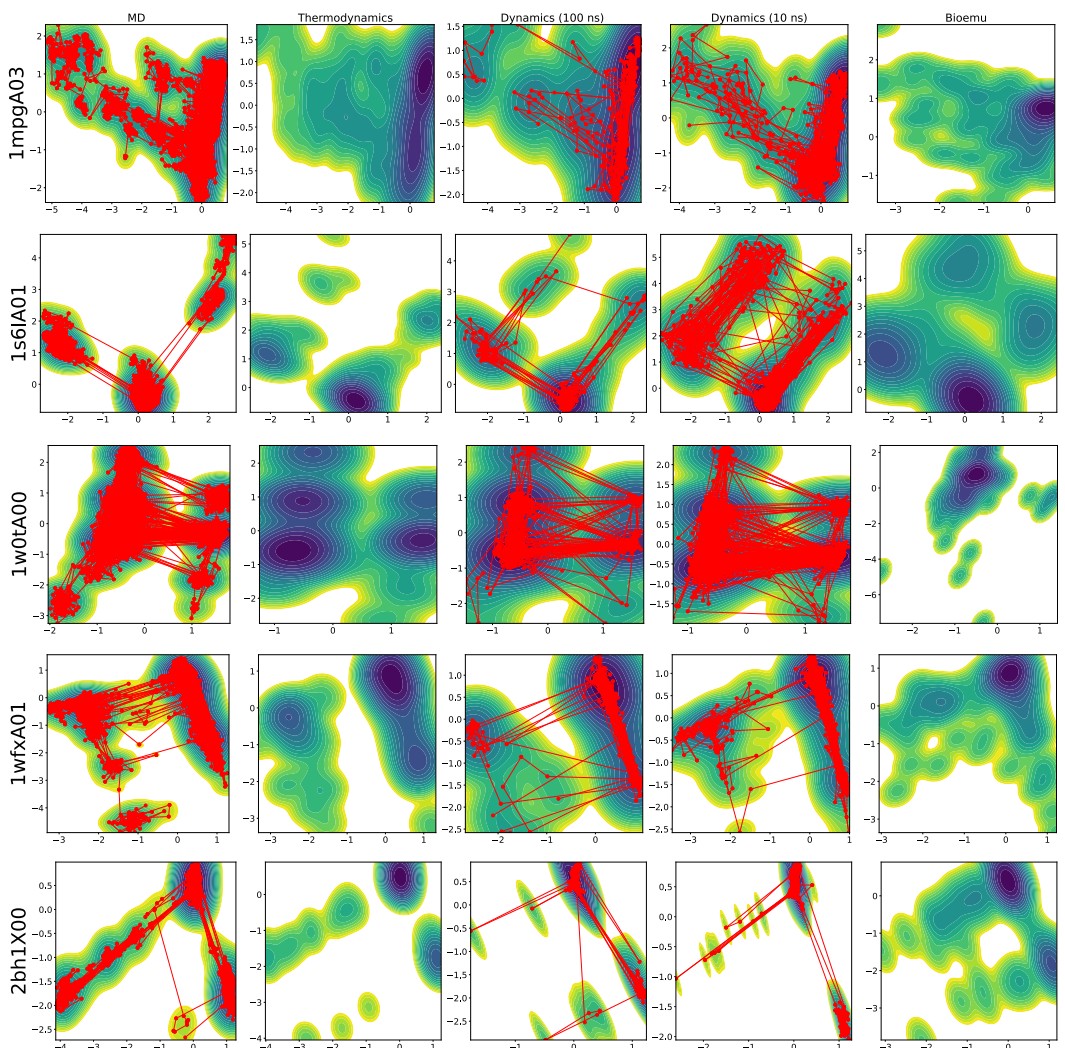

Figure 6: Additional results on another 5 test CATH1 proteins.

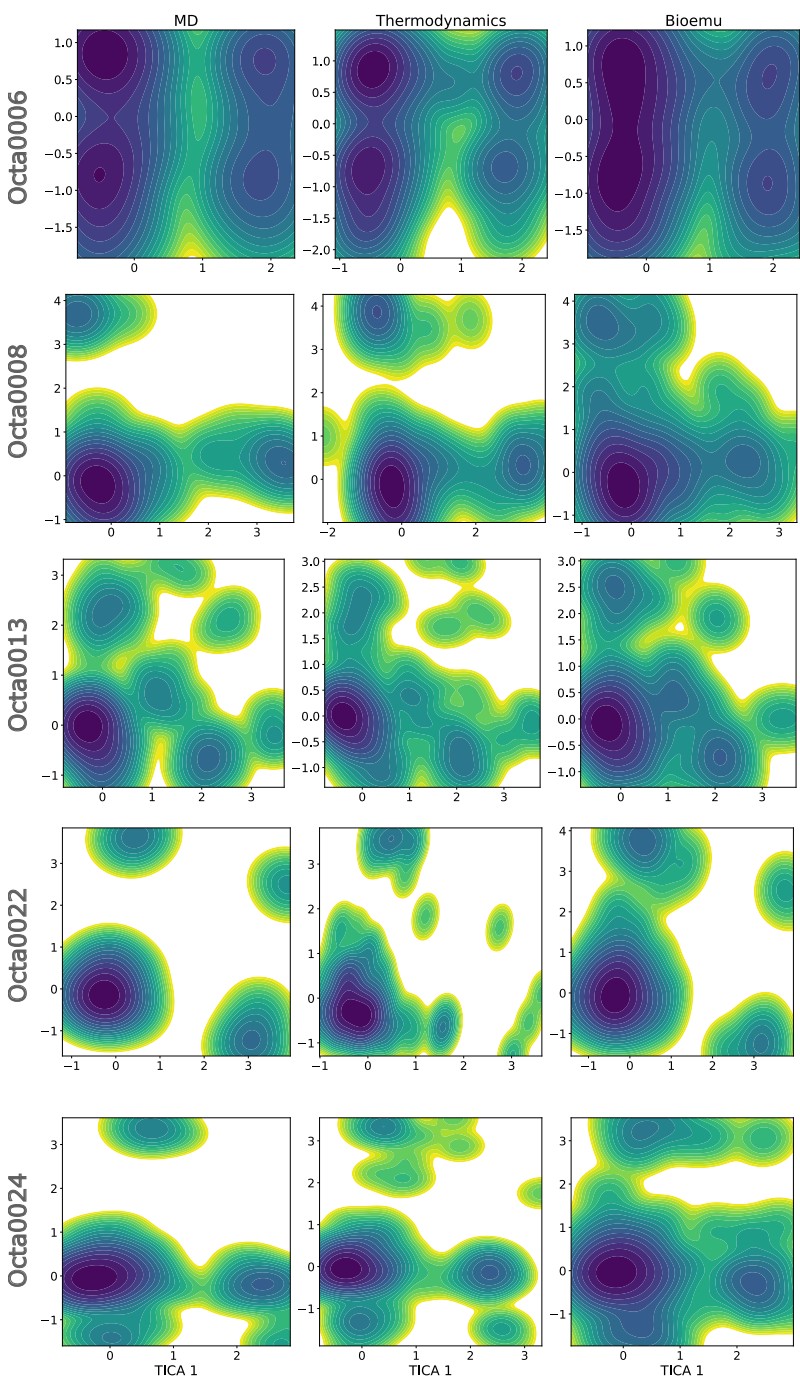

Figure 7: Additional results on 5 test Octapeptides: free energy surface along the top two TICA components.

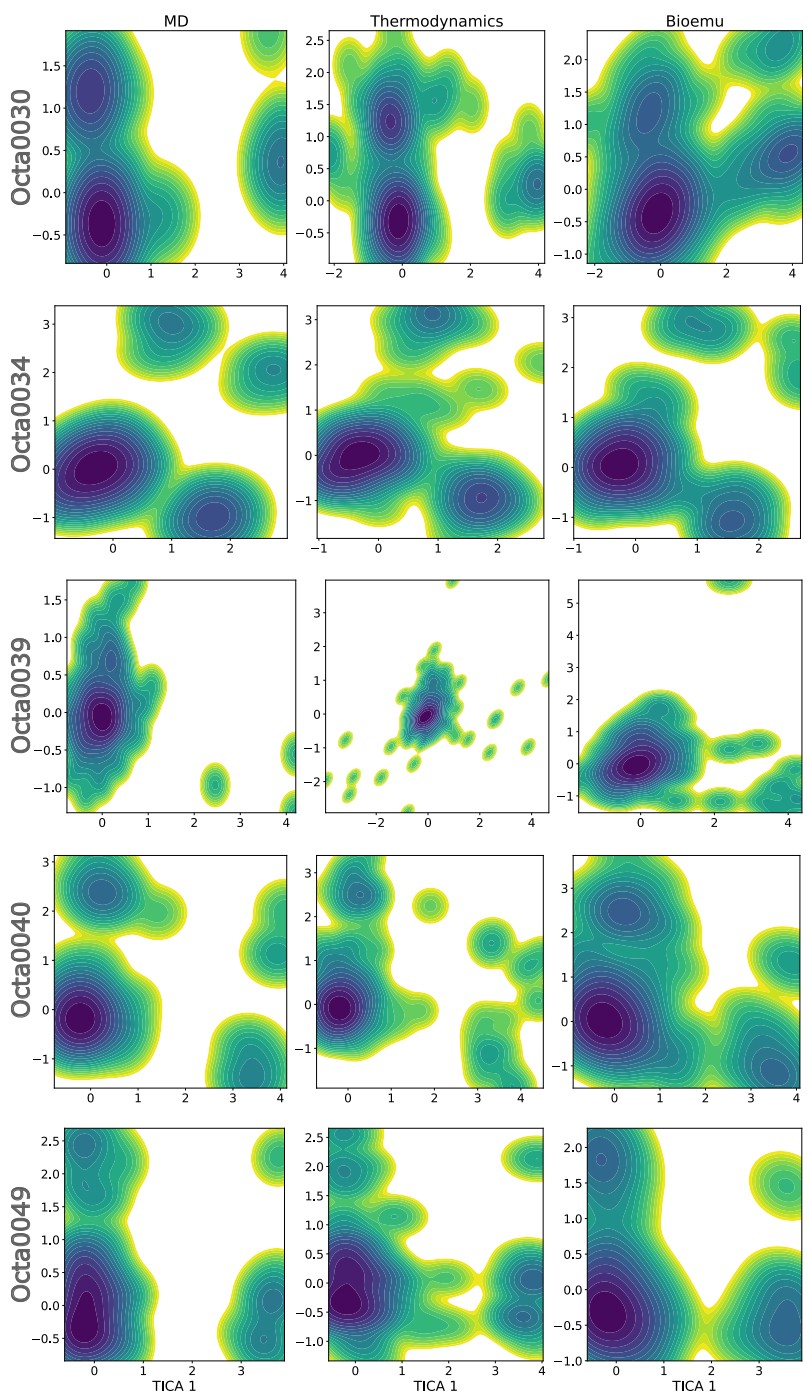

Figure 8: Additional results on another 5 test Octapeptides: free energy surface along the top two TICA components.

