# OpenReview forum: "ProTDyn: A Foundation Protein Language Model for Thermodynamics and Dynamics Generation"
_ICLR.cc/2026/Conference — ICLR 2026 Poster_

### Official Review · Reviewer_x8Ec · 2025-10-29

**Soundness:** 2
**Presentation:** 2
**Contribution:** 2
**Rating:** 6
**Confidence:** 2

**Summary:**

The work proposes ProTDyn, a model that is able to do conformation ensemble generation and multi-timescale dynamics modeling within a single model.

**Strengths:**

The paper is well-written and easy to follow. The fact that the authors present a model that is able to perform simultaneously equilibrium conformation ensemble generation and multi-timescale dynamic trajectory generation is interesting and as far as I am aware novel.

**Weaknesses:**

As the paper is not near my main area of research, it is hard for me to make questions or comment on the weaknesses although I found the paper easy to follow and appreciated especially the explanation on page 4.

The paper is a rather standard application of ML modelling techniques but unfortunately I cannot comment on the strength of the results. For this reason I will leave a weaker confidence score.

**Questions:**

I'm slightly confused by Figure 1 as the authors mention it is a language model, but there does not seem to be language/text as input in the diagram. Could the authors please clarify? Are the authors using "language model" to highlight the fact that it is an autoregressive (causal/decoder) Transformer architecture and not that the input is language.

---

> ### Author Response · Authors · 2025-11-23
> **New dynamics-only baseline model trained and other questions answered**
>
> Thank you for the interest! Protein biophysics is indeed interesting from both a physical and a machine-learning perspective.
>
> **Question-1**: We use a standard protein language–modeling formulation, where each amino acid in the sequence is represented as a sequence token, and the corresponding amino-acid structure is encoded as a structure token. This provides a tokenized representation for both sequence and conformation, allowing the model to operate within a unified language-model framework.

---

> > ### Comment · Reviewer_x8Ec · 2025-11-27
> >
> > Thank you for the response. I have read the other rebuttals and I am happy to keep my score. I am unfortunately going to keep my low confidence as it is distant from my main area of research.

---

### Official Review · Reviewer_7aKu · 2025-10-30

**Soundness:** 2
**Presentation:** 2
**Contribution:** 3
**Rating:** 6
**Confidence:** 3

**Summary:**

In this paper, the authors present ProTDyn which is a transformer operating on ESM3 structure tokens trained for conformation and dynamics sampling. The model is trained on a mixture of the alphafold synthetic database of structures and molecular dynamic datasets. It is trained for both equilibrium distribution sampling as well as jumping ahead in MD simulations or inpainting in between coarse time steps in MD simulations. The model improves over BioEmu, a recent deep learning model for conformer generation, in terms of distributional match to MD.

**Strengths:**

The training of the model is simple in that all 3 tasks of thermodynamics, dynamics and dynamics inpainting are trained jointly and simultaneously in a single model. This avoids the need for any task specific fine-tuning which is a downside of prior approaches. All tasks are unified in terms of autoregressive modelling with different factorizations.

The experimental results look strong in terms of improvement over the BioEmu baseline model. Distibutional fit on MD test data looks quite convincingly better in Table 1 and in Figure 2.

The authors make some interesting observations in terms of the time resolution with which to sample their dynamics model. They find that sampling with a finer grid can lead to worse overall performance potentially due to error accumulation. This finding will be useful for further research into deep learning based dynamics models.

**Weaknesses:**

A main weakness of the paper is the lack of ablations with respect to the stated contribution of ProTDyn being a unified model for Thermodynamics and Dynamics generation. On L56-L58 in the introduction the authors describe how joint training can be mutually beneficial for both tasks. I would therefore expect experiments and ablations showing that this is indeed the case. You could train a version of your model on the tasks individually and compare performance with the multi-task trained model. I don't think comparing to BioEmu is enough to make this claim since BioEmu is pretrained on Thermodynamics style tasks and fine-tuned on dynamics tasks which can be seen as a form of multitask training.

With regards to Table 2, where these proteins are in your test set but BioEmu's training set, I am unsure how ProTDyn ended up with a different test set to BioEmu that made this comparison difficult. If you are using the same datasets as BioEmu, why not also use the same splits? In the end, it makes it very hard to draw any conclusions from Table 2 since the models are not comparable.

The paper is quite unclear with the usage of the term 'module'. The authors state that they have a 'thermodynamics module' and a 'dynamics module'. This would imply separate parts of the network that are being trained for these two tasks which would be in contradiction with the introduction describing a single unified model which is a key stated contribution of the work. I am unsure what the authors are referring to with regards to these modules and this should be made clearer.

Further, in terms of clarity, on L149 the authors describe their model as a 'multimodal protein language model'. In what way is this a multimodal model because as far as I am aware the model just generates structural tokens (and no other modalities).


I am tending to accept this paper due to the strong experimental results with regards to BioEmu and simple training scheme. This model seems to make a significant improvement on the state of the art for conformational modelling.

**Questions:**

Why did you decide to use a causal autoregressive distribution for the generation of structural tokens? In ESM3, an any-order autoregressive model was used to generate tokens in parallel which seems to make sense for protein data. Did you try any other parameterizations?

How were the proteins in Figure 2 selected? If these were cherry-picked as to best show the benefits of your method this should be stated clearly. Or if these are exhaustive of the CATH2 dataset, this should also be made clear because this is important for judging the significance of your findings.

---

> ### Author Response · Authors · 2025-11-23
> **New dynamics-only baseline model trained and other questions answered**
>
> Thank you for the reviewer’s detailed and thoughtful discussion. We have uploaded a revised version of the paper, with all updates highlighted in blue. The major change is the addition of a newly trained dynamics-only baseline model, trained using the same MD simulation dataset (the single-structure dataset is only relevant for thermodynamics and is therefore excluded).
>
> **Weakness-1**: We now include a dynamics-only baseline. As shown in Table 1 and Table 3, removing thermodynamic supervision causes the model to fail to generate physically plausible trajectories. The dynamics-only model rapidly diverges from the folded structure and performs poorly in reproducing realistic stationary distributions and transition statistics. These results demonstrate the importance of jointly learning thermodynamics for stabilizing the predicted dynamics. Intuitively speaking, understanding thermodynamics helps the model “know” what plausible structure that can be generated, thus avoiding generating implausible structures to a sequence. BioEmu itself is a thermodynamics-only model and I am not aware if it has a recent dynamic version.
>
> **Weakness-2**: The Octapeptide dataset is the only dataset in the Microsoft datasets that enables quantitative comparison of ensemble quality using collective-variable energy landscapes. It contains many short proteins of identical length (8 residues), making quantitative generalization possible. BioEmu does not separate this dataset into training and testing, whereas our goal is to evaluate generalization to unseen proteins. The purpose of this comparison is to demonstrate that our model can generate ensembles for unseen sequences that closely match those produced by a state-of-the-art ensemble generator such as BioEmu (even if ensembles of these proteins have been seen by BioEmu).
>
> **Weakness-3**: I thank the reviewer for pointing out the misuse of terminology. The thermodynamics and dynamics components are not separate modules but rather separate prediction heads on top of a shared backbone. We have replaced the term “module” with “head” throughout the paper for clarity.
>
> **Weakness-4**: We have replaced the term "multimodal" with "multi-task" to avoid confusion, since all tasks operate on the same structural-token representation.
>
> **Question-1**: I agree that an autoregressive model is not necessarily the most expressive model in principle. However, any-order training (as used in ESM-3) becomes substantially more expensive when applied to simulation-based datasets, where each protein sequence may correspond to more than a thousand structures. This makes any-order training impractical at the current scale. Recent work has explored adapting ESM-3 to masked-diffusion language training [Lu, Jiarui, et al. "Structure language models for protein conformation generation." arXiv preprint arXiv:2410.18403 (2024).], but it remains unclear whether the advantages of masked-diffusion language models outweigh those of continuous 3D-structure diffusion models. At present, the autoregressive formulation is both tractable to train and able to output explicit probabilities, which is important and non-trivial for diffusion-based models.
>
> **Question-2**: The example shown in the main text was randomly selected, not cherry-picked. To further address this concern, there were more visualization results in Appendix Figures 5–8.

---

### Official Review · Reviewer_NTbS · 2025-10-31

**Soundness:** 4
**Presentation:** 4
**Contribution:** 3
**Rating:** 8
**Confidence:** 5

**Summary:**

This paper introduces ProTDyn, a foundation PLM to unify equilibrium distribution sampling, time-dependent dynamics sampling as well as inpainting for fine-grained timesteps based on coarse grained timesteps. It has been trained on a large scale molecular dynamics dataset, and demonstrated promising performance in the generated sample distribution in comparison with baseline models. I find this methodology very interesting and novel, and has shown scaling up in the model and data.

**Strengths:**

**Novel methodology**
- The method is novel to incorporate three tasks in one model:
  1. i.i.d. equilibrium distribution sampling
  2. time lagged predictions with multiple time scales
  3. inpainting: given two coarse grained time steps, predict fine grained steps in between
- This methodology overcomes the lack of capability in the description of dynamics/kinetics in the equilibrium sampling models

**Transferability**
- The model is trained on multiple time scales, showing transferability in the time dependence
- The model is trained on a large scale of MD data, showing transferability in the chemical space

**Performance**
- The performance has been compared on MD sample distributions with proper baselines and benchmarks, and has demonstrated the advantage of this model

**Weaknesses:**

- **Generalization**: For the dynamics tasks, the test proteins were seen by the model during thermodynamics training, which may not represent true generalization. A stronger evaluation would involve holding out proteins based on sequence similarity
- **Baseline**: This model is only compared against BioEmu as the only baseline. It can consider adding more baseline models, such as Alphaflow which is also trained on mdCATH, as well as a few other works that baked in the time dependence for dynamics generation of protein models.
- **Inpainting task**: Dynamics inpainting is highlighted as a key capability, but its performance is only measured indirectly through the quality of the final, end-to-end generated trajectories. There are no specific benchmarks that isolate and quantify the accuracy of the inpainting process itself

**Questions:**

- Could you clarify the train/test splitting procedure? Was any filtering based on sequence identity or structural similarity
- Have the authors considered other architectures such as those AF-based models? What was the specific rationale for choosing the ESM3 architecture?
- BioEmu has been finetuned on experimental data after training on MD, so the comparison on MD distribution might not be the most direct comparison, which can be noted.
- The method aims to generate coarse-grained dynamics. It'll be helpful to demonstrate the computational cost in comparison with MD simulation
- Have the authors considered the asymptotic behavior of the time dependence? e.g. when the $\delta t \to \inf$, it should approach an i.i.d. distribution

---

> ### Author Response · Authors · 2025-11-27
>
> Thank you for the reviewer’s detailed and thoughtful discussion. We have uploaded a revised version of the paper, with all updates highlighted in blue. The major change is the addition of a newly trained dynamics-only baseline model, trained using the same MD simulation dataset (the single-structure dataset is only relevant for thermodynamics and is therefore excluded). I apologize for the delayed response as many of your points are deep and take me some actual experiments to verify.
>
> **Weakness-1**: We agree that true generalization would require quantitatively accurate dynamics for completely unseen proteins, which is likely beyond the capability of the current MD simulation datasets and purely data-driven approaches. Our generalization claim is therefore intentionally weaker: once the thermodynamic landscape is known, the model can recover quantitatively correct dynamical behavior. This was the motivation for unifying thermodynamics and dynamics within a single framework. As confirmed by our new ablation experiment, a dynamics-only model fails to recover the correct dynamics, whereas the full model—augmented with thermodynamic information—succeeds.
>
> **Weakness-2**: We acknowledge prior work such as AlphaFlow and Str2Str. However, most earlier models rely either on single-structure-only training or on very limited MD datasets (typically only hundreds of nanoseconds). BioEmu is, to our knowledge, the first method—and still the state of the art—that is trained on both large-scale single-structure data and high-quality long-time MD trajectories. Our own model is trained on a dataset of similar scale and quality. For this reason, we consider BioEmu the most appropriate and fair baseline for comparison.
>
> **Weakness-3**: Transition path sampling (TPS) is indeed an important topic in computational chemistry. Our third module is therefore phrased as dynamic inpainting rather than transition-path sampling, and we do not evaluate its performance. This is because the current general-purpose MD datasets may not reliably capture high-energy transition-state ensembles, making them suboptimal for quantitative TPS training and evaluation. We have added a brief discussion of this limitation in Lines 490–494.
>
> **Question-1**: For the thermodynamic generalization experiment on the Octapeptide dataset, we use random 99%–1% train–test splitting on sequence. For the dynamics experiment, the entire CATH1 dataset is used strictly as dynamics test data, with no sequence overlap with the training set.
>
> **Question-2**: For future extensions of this work, we want an explicit likelihood (very important for introducing any energy function or statistical mechanics principles), even if defined in a tokenized rather than continuous space. Diffusion models are powerful but do not permit tractable and efficient likelihood evaluation in their standard form. To enable this capability, we adopt the ESM3-style framework.
>
> **Question-3**: This is very fair point! The RL finetuning stage of BioEmu does shift the distribution.
>
> **Question-5**: This is an important point. As Δt → ∞, the dynamic prediction should indeed reduce to the thermodynamic (equilibrium) distribution. Our current implementation does not explicitly enforce this constraint. A straightforward method would be to at inference time, introduce a Δt threshold above which the temporal embedding is set to zero and cross-segment attention is disabled, causing the model to collapse naturally into the thermodynamic limit.

---

### Official Review · Reviewer_87Fq · 2025-10-31

**Soundness:** 3
**Presentation:** 2
**Contribution:** 2
**Rating:** 4
**Confidence:** 3

**Summary:**

The paper proposes a protein foundation model based on ESM3 that unifies equilibrium distribution modelling and dynamics generation for protein structures. Training of the model optimizes the loss for distribution modeling, dynamic trajectory prediction, and dynamic trajectory inpainting at the same time. To enable the modeling of dynamics, ProTDyn proposes two-layer rotary embedding scheme to combine time rotary embedding. Experiment results show that ProTDyn performs than baselines in metrics including distributional similarity. ProTDyn also shows reasonable performance in capturing state transitions consistent with molecular dynamics.

**Strengths:**

1.The paper proposes an interesting view of enabling one model for both equilibrium distribution sampling and temporal molecular dynamic modeling.

2.The experiment results prove the effectiveness of the method in distribution modeling compared with strong baselines.

3.The paper is clearly written with the mothod being simple yet elegant.

**Weaknesses:**

1.The benefit of unifying thermodynamics and dynamics generation is not explicitly discussed in this paper. It would be interesting to see how the two tasks interfered with each other in terms of performance. This would also provide stronger support for the motivation of the unification.

2.Some important details of the method are missing. For example, how the “two-layer rotary embedding scheme” is designed to combine temporal and residue positions. Given the model architecture leverages the ESM3 backbone, such details are important for the readers to understand the key adaptation to enabling the dynamics generation.

3.The evaluation metric for the distributional similarity are along low dimensional collective variables. Specifically, distribution along the Rg and RMSD (which are single values not reflecting details of the protein structures) w.r.t. the native structure seems to be less significant in reflecting the true distribution in 3D Euclidean space with all the residues.

**Questions:**

1.Before ProTDyn, the paper “Two for One: Diffusion Models and Force Fields for Coarse-Grained Molecular Dynamics” also discusses how the connections between score-based generative models and force fields can be leverage to train a diffusion model on MD simulations, and the trained score function can be used to simulate MD trajectories. It would be valuable to clarify the difference and advantage of ProTDyn when compared with these methods.

2.It would be interesting to see how ProTDyn works on larger proteins. Particularly, it would both enhance the presentation of the paper and provide more evidence of the generalizability of ProTDyn if the predicted structure/distribution can be demonstrated with figures.

---

> ### Author Response · Authors · 2025-11-23
> **New dynamics-only baseline model trained and other questions answered**
>
> Thank you for the reviewer’s detailed and thoughtful discussion. We have uploaded a revised version of the paper, with all updates highlighted in blue. The major change is the addition of a newly trained dynamics-only baseline model, trained using the same MD simulation dataset (the single-structure dataset is only relevant for thermodynamics and is therefore excluded).
>
> **Weakness-1**: We now include a dynamics-only baseline. As shown in Table 1 and Table 3, removing thermodynamic supervision causes the model to fail to generate physically plausible trajectories. The dynamics-only model rapidly diverges from the folded structure and performs poorly in reproducing realistic stationary distributions and transition statistics. These results demonstrate the importance of jointly learning thermodynamics for stabilizing the predicted dynamics. Intuitively speaking, understanding thermodynamics helps the model “know” what plausible structure that can be generated, thus avoiding generating implausible structures to a sequence.
>
> **Weakness-2**: We have added a detailed description of our rotary embedding scheme, including both residue-position embeddings and temporal embeddings, in L227–237.
>
> **Weakness-3**: We believe that combining physical feature analysis (radius of gyration, RMSD relative to the folded structure) with TICA provides a strong and complementary evaluation.
> - The radius of gyration quantifies structural compactness.
> - The RMSD relative to the folded state reflects thermodynamic stability.
> - TICA examines whether the model captures the correct metastable states and slow collective modes of the underlying energy landscape.
>
> Together, these analyses provide a comprehensive evaluation of both structural and kinetic behavior.
>
> **Question-1**: We added a sentence in L41–42 explaining what is missing in current approaches that attempt to connect thermodynamics and dynamics. Specifically, for the CG model referenced by the reviewer:
>
> - Even with accurate bottom-up coarse-graining, the CG simulation does not guarantee recovery of the correct dynamics.
> - The method uses the score at small diffusion times as an effective force for propagating an underdamped Langevin process; recent work [Plainer, Michael, et al. "Consistent sampling and simulation: Molecular dynamics with energy-based diffusion models." arXiv preprint arXiv:2506.17139 (2025).] has shown that this approximation is insufficient.
> - The resulting force-based propagation requires a small integration timestep, which inevitably leads to cumulative numerical error.
>
> Our approach avoids these limitations by learning thermodynamics and dynamics jointly in a single model without explicit simulation-based propagation.
>
> **Question-2**: We agree that generalizing to large proteins (and intrinsically disordered proteins) is an important direction. However, this is beyond the realistic scope of current models and datasets. Capturing global motions of large proteins requires long MD trajectories that are often beyond current MD simulation capabilities, making both training and evaluation challenging. Extending the model to these systems is an exciting goal for future work.

---

> ### Comment · Reviewer_87Fq · 2025-11-27
>
> The additional results in Table 3 that a dynamic-only model shows degraded dynamic performance do partly address my concern about the motivation of combining dynamic and equilibrium.
>
> And I see the potential adavantage of the proposed approach over force-based simulation. It could be emphasized more if there's evidence that cumulative numerical error could be reduced in the token-wise autoregressive approach, which will be an important highlight of the paper. But experiments providing such evidence could be difficult.
>
> In addition, I think Figure 1 could be further improved for better presentation in the camera-ready version if the paper is accepted.
>
> I will raise my score to 6 given the authors' response.

---

### Meta-Review · Area_Chair_RRjd · 2026-01-04

**Summary:**

This paper introduces ProTDyn, a unified protein language foundation model with 1.4B parameters that jointly performs equilibrium ensemble generation, multi-timescale dynamics prediction, and dynamics inpainting. ProTDyn is trained on a large-scale protein dynamics dataset and demonstrates superior performance over the baseline BioEmu.

Strengths:
(1) The model unifies thermodynamics, dynamics, and dynamics inpainting within an autoregressive framework. (2) The model is trained across multiple timescales, making the proposed approach flexible and broadly applicable. (3) ProTDyn consistently outperforms the baseline BioEmu across reported experiments.

Weaknesses:
(1) A key limitation is the lack of ablation for the benefit of jointly modeling thermodynamics and dynamics, which is critical to support the paper’s motivation (Reviewer 87Fq, Reviewer 7aKu).
(2) The presentation of the paper shoud be improved, as some implementation details and experimental settings are not described with sufficient clarity (Reviewer 87Fq, Reviewer NTbS, Reviewer 7aKu).
(3) The experiment relies on limited baseline (only bioEmu), and there are concerns regarding dataset splits and evaluation protocols (Reviewer NTbS, Reviewer 7aKu).

**Reviewer Concerns:**

The authors provided detailed responses to all comments of four reviewers, addressing most of the core concerns.

Reviewer 87Fq primarily questioned the advantages of jointly modeling thermodynamics and dynamics. The authors directly addressed all three weaknesses raised by this reviewer with reasonable and technically sound responses。

For Reviewers NTbS and 7aKu, the rebuttal mainly consisted of clarifications and methodological justifications, further explaining the design choices.

(2) The authors added the necessary implementation details and experimental details in response to the reviewers’ comments and revised the manuscript accordingly. However, several aspects of the manuscript remain insufficiently explained. The authors should further refine the presentation of the paper and enhance the clarity and readability of the manuscript.

**Reviewer Scores:**

Given the main concerns addressed, it is likely that Reviewer 87Fq would increase their score.

For Reviewers NTbS and 7aKu who already assigned relatively positive scores (with initial scores of 8 and 6, respectively)), a substantial score increase appears less likely.

Reviewer x8Ec explicitly noted that the paper falls outside his main area of research. I assign a relatively low weight to this reviewer’s evaluation when forming my final recommendation.

The first three reviewers expressed positive assessments of the paper’s contributions and practical value. Taking all reviews and rebuttal discussions into account, I recommend that this paper be accepted.

---

### Decision · Program_Chairs · 2026-01-26

Accept (Poster)